# Cultural Consumption and Knowledge, Attitudes, and Practices Regarding Waste Separation Management in China

**DOI:** 10.3390/ijerph19010338

**Published:** 2021-12-29

**Authors:** Aiqin Wang, Sijia Dang, Wenying Luo, Kangyuan Ji

**Affiliations:** School of Economics and Finance, Xi’an Jiaotong University, Xi’an 710061, China; dang2556@stu.xjtu.edu.cn (S.D.); ireneluo2000@163.com (W.L.); yl1013@stu.xjtu.edu.cn (K.J.)

**Keywords:** cultural consumption, waste separation management, knowledge, attitudes and practices

## Abstract

In 2017, the Chinese government created a policy on mandatory waste separation. Many communities and cities have created waste management institutions and appointed workers to supervise these actions. But there is little information about the situation in terms of the knowledge, attitudes, and practices of waste separation and any differences among regions and cities. Thus, the goal of this paper is to show the current status quo and any differences and to analyze their determinants, especially regarding cultural consumption. Based on online survey data collected in 2021, we found that knowledge in rural regions was lower than in urban regions, but there was no difference in attitudes or practices; the practices in pilot cities were better than in non-pilot cities, but the knowledge and attitudes showed no differences. Different cultural consumption patterns had different impacts on waste separation knowledge, attitudes, and practices. Based on the results, a policy related to culture should be enacted to improve efficiency and increase the action impacts to solve environmental and social issues.

## 1. Introduction

After China’s reform and opening up in 1978, great changes have taken place in China, especially in the daily lives of ordinary people. But the waste quantity generated with the improvement in life conditions is increasing, and waste composition and categories are becoming more complex; thus, separating waste at the source, in the house and community, is the most fundamental solution for waste management, and it has also become an urgent issue in trying to lower the cost of waste management processes and services [1,2,3]. Governments around the have world enacted policies and implemented activities to improve their populations’ waste separating behavior; for example, there are voluntary environmental programs, unit-charging programs, waste wars in Japan, and producer-pay institutions in Germany [4,5,6,7,8]. In China, the government has also established several institutions and policies to change waste separation services such as the pilot waste source-separated collection program, which started in eight cities in 2000, and the solid waste mandatory separation plan in approximately 46 cities in 2017 [9,10]. In addition, the Chinese government made waste sorting an element of the Revised Solid Waste Law in April 2020. But the consequences of these policies are comparatively worse in some regions, and on the level of individuals and communities, the reasons for this include low waste separation knowledge and attitudes, lack of time to engage in waste separation, no incentives or punishment mechanisms, inadequate separated-waste collection vehicles, a lack of good and efficient advertisements, and a lack of comprehensive waste management services (which cause inefficient waste disposal without waste classification, no legal constraints, and classification at the source and mixing during the transportation) [11,12,13,14]. However, the essential and intrinsic reason is the relatively low motivation for waste separation among residents with regard to community waste management services, which improves waste separation knowledge and attitudes and increases the possibility of related behavior [15].

Many studies on knowledge, attitudes, and practices (KAPs) regarding waste management, including waste separation and recycling, have mainly focused on their relationships and determinants based on the theory of planned behavior. The impacting factors included past experiences, motivation, social morals, intentions, facilities’ conditions, perceived values, and socioeconomic characteristics [16,17,18,19,20,21]. In addition, KAP relationships are comprehensive and inter-impacted [22,23,24,25,26]. In China, previous studies found that knowledge and behavior regarding waste separation were low but attitude was higher. There is little evidence on the KAPs of residents concerning waste separation collection across difference regions, especially in China and after the pilot city regulation of waste separation. Cultural consumption is a spirit of consumption that increases social knowledge, improving attitudes toward social phenomena, and changing behavior. Some researches, using Italian municipal data, have stated that cultural consumption can improve behavior toward waste recycling, especially newspapers, cinemas, museums, and books can all increase waste recycling behavior [27]. In the literature and practice of waste management, there is little evidence that KAPs or their motivation are caused by cultural consumption in China.

Based on motivation theory, motivation can be divided into intrinsic motivation and extrinsic motivation [28]. Intrinsic motivations can incentivize a person to take the initiative to adopt behavior for internal rewards and altruistic values, but extrinsic motivation pushes a person to improve behavior for external rewards and outside pressures (such as material incentives and social appraisal) [29,30]. Intrinsic motivation is important as a dominant factor in the effectiveness of external conditions [31]. Several studies have proposed that extrinsic motivation does not cause a stronger and more durable influence on behavior but that intrinsic motivation is the main driver of individual behavior [32,33] and can promote long-term pro-environmental behavior without continuous monetary benefits [34,35]. For example, intrinsic motivation can decrease alcohol consumption over the long term [36]. Extrinsic and intrinsic motivation can impact KAPs toward waste management. Extrinsic motivation can increase the effort level of enhancing knowledge and attitudes, while intrinsic motivation can steadily increase and maintain waste separation behavior to a certain degree [37,38,39,40,41].

Cultural consumption is a choice of leisure activities based on an individual’s social motivations, values, and learning by consuming goods [42,43]. It is also an important instrument for shaping and improving an individual’s value structure and collective identity during the consuming process [44], driving forces that impact waste separation practices and behavior [45,46,47,48,49]. In the art world, cultural goods are broadly classified as either highbrow or popular art and culture [50,51]. Those with elite artistic taste pursue highbrow art such as classic music and drama; on the contrary, those with popular artistic preferences usually engage in contemporary or individualist contexts using the masses’ expressions [52]. Mass cultural consumption results in materialism and causes residents to pursue external motivations and display indifference to social questions; this decreases community participation and environmental intentions [12,53]. Movies are a typical example of materialism; they are always within a context that encourages individuals to chase material or status success and individual happiness [54]. However, elite cultural consumption leads to awareness and altruism and increases intrinsic motivation for action, promoting the pro-environmental, pro-social knowledge, and attitudes needed to be converted into behavior [12]. For example, visiting a museum can help residents learn about history and reinforce social participation and identity [55]. Thus, cultural access can promote residents’ waste separation collection KAPs through the process of consuming cultural goods, such as reading books, watching movies, and listening to musical concerts, that are focused on environmental or related issues.

We first describe and compare the different impacts of cultural consumption on KAPs regarding waste management among different regions in China. Thus, the specific goal of this paper was to (1) describe the KAPs of waste separation in China and differences among rural and urban regions as well as in pilot waste separation cities and non-pilot waste separation cities; (2) describe and analyze their relationships and the impact of cultural consumption on KAPs. The structure of the paper is as follows: Section 2 presents the sample, model, and methods as well as the variables and their characteristics. Section 3 provides the results, including the situation of KAPs regarding waste separation, the relationships between cultural consumption and KAPs, and their impacts. Section 4 offers a discussion that explains the results and their possible reasons. Section 5 is the conclusion, which discusses the implications of the results and suggestions for future research.

## 2. Data and Methods

### 2.1. Data Collection

We designed a questionnaire using Jinshuju, developed by ThoughtWorks, which is an application program that designs and sends out questionnaires, and we conducted a national online survey about solid waste management services using WeChat between October 2020 and January 2021. Our research team included six undergraduates and two graduate students, who were seed investigators. We trained them to help participants complete the survey. The inclusion criteria for participants were (1) 18 years old or above; (2) completion of the survey using WeChat; (3) only submitting answers through the same IP address once, which was restrained by Jinshuju; (4) voluntary participation. Before the formal investigation, we conducted a pilot survey to test the questionnaire’s rationality and validity. After the survey, the survey randomly sent a small remuneration to the participants as an incentive to increase the response rate, which was on average USD 0.16 dollar (i.e., RMB 1 in China), and ninety percent received the remuneration [37]. In total, 1200 participants completed the questionnaire during 4–10 February 2021, which was one week before the Spring Festival, ensuring that the information collected was correct within a particular context. In addition, 1189 samples were validated with integral socioeconomic information and the KAPs of waste separation, and participants were from 185 cities in 31 provincial areas in China (Table 1). We divided the cities into waste separation pilot cities and non-pilot cities based on the filled-in addresses and national documents.

In order to explore the impact of cultural consumption on knowledge, attitudes, and behaviors regarding waste separation, we first calculated the average knowledge score on how to separate recyclable waste and harmful waste using the right way, resulting in a validity score of 0.6. To compare the differences among the regions, we simply used the same recycling and harmful waste classification based on local regulations and practices (The central government only has institutions related to waste separation collection in urban regions, but local governments have specific separation collection methods, and there are many differences among cities. However, the points related to recycled waste and harmful waste are the same. Thus, we used these two knowledge scores as the waste knowledge. Basically, recycled waste includes national glass containers, plastic bottles, scrap metal, and old books or newspapers; harmful waste includes used batteries, medicine bottles, and used syringes and needles.) Next, we calculated the average attitude scores using willingness to separate waste in different situations (including without separation facilities and with house and community waste separation facilities), participating in community waste activities, and paying for waste services, with a validity of 0.9; we then used daily waste separation behavior as waste practice. We also asked residents about the difficulty of separating waste and their waste separation habits. Then, we surveyed their cultural consumption including news, books, cinemas, museums, dramas, and cultural sites. We next collected basic information on the residents including gender, age, education level, monthly income, health, and environmental activities. Finally, we asked about community waste services including waste separation advertisement, rules, and facilities. These questions and answers are shown in Appendix A Table A1.

### 2.2. Model and Methods

Because of the motivation of cultural consumption and the forming process of knowledge, attitudes, and practices of waste separation, we used the theory of planned behavior and motivation theory in combination with descriptive and multivariate analysis methods to analyze the impacts of cultural consumption on waste separation knowledge, attitudes, and practices. Based on the relationship of KAPs, we created an ordinary least squares (OLS) regression model to analyze the impacts of cultural consumption on waste separation KAPs with the following equations separately. Based on Bourdieu’s cultural sociology theory and Karl’s consumerism sociology theory [56,57], mass and elite cultural consumption broadly represents two different individuals’ values and motivations (i.e., materialism and altruism), and we separated cultural consumption into mass cultural consumption and elite cultural consumption, which were added to the model separately to analyze their impacts.
K_i_ = α + β_1_ Cul_i_ + β_2_ In_i_ + γX_i_ + μR_i_ + ε_i_,(1)
A_i_ = α + φ_1_ K_i_ + β_1_ Cul_i_ + β_2_ In_i_ + γX_i_ + μR_i_ + ε_i_,(2)
P_i_ = α + φ_1_ K_i_ + φ_2_ A_i_ + β_1_ Cul_i_ + β_2_ In_i_ + γX_i_ + μR_i_ + ε_i_,(3)
where K_i_, A_i_, and P_i_ indicate the waste separation knowledge, attitudes, and behavior, respectively, of residents i. K_i_ is the average knowledge score regarding recycling waste and harmful waste, which are continuous variables. A_i_ is the average willingness to engage in waste separation and community participation, which is also a continuous variable. P_i_ is the behavior of residents i with regard to separating waste in their daily lives (when the resident separates waste, P_i_ is equal to 1; otherwise, it is equal to 0). Cul_i_ is cultural consumption including news, books, movies, museums, dramas, and cultural sites. Mass cultural consumption included news, books, and movies, and elite cultural consumption included attending museums, dramas, and cultural sites. In_i_ are the community facilities and institutions of waste management, which include waste separation facilities, rules, and advertisements. R_i_ are a region’s characteristics, which are dummy variables for controlling for community location effects, which include urban and rural regions, waste separation pilot cities, and non-pilot cities. X_i_ is a vector of residents’ characteristics including age, education, gender, monthly household income, health, and environmental activity. The symbol α is a constant term, and the symbols φ_1_, φ_2_, β_1_, β_2_, μ, and γ are the coefficients to be estimated. ε_i_ is the error term.

Then, we used the probit model for testing robustness, as shown in Appendix A Table A2, to show the effects of cultural consumption on the difficulty of separating waste and the habit of conducting more testing of the results’ robustness. The difficulty of separating waste is rooted in the difficulty in engaging in waste separation behavior for a variety of reasons, which is a binary variable.

### 2.3. Sample Characteristics

The descriptive statistics of socioeconomic characteristics, community service provision for waste separation, and community location are presented in Table 1. More than 57% of residents were women, and two out of three were younger than 24 years old. Most participants’ education level was a bachelor’s degree. Their income was relatively low, and 44% of participants earned less than USD 782 per month. The health situation of sixty-nine percent of residents was good or better. In addition, 30% of residents took part in environmental activities last year. Regarding their communities, residents reported that 64% of communities had waste separation advertisements, 56% of communities had waste rules, and only 28% had waste separation facilities. In total, 70% of residents lived in urban regions, and 49% of these lived in a pilot waste separation city.

## 3. Results

### 3.1. Knowledge, Attitudes, and Practices of Waste Separation and Cultural Consumption in China

In Table 2, we first report the KAPs of waste separation in detail. Twenty-six percent of residents had proper waste separation knowledge, 83% residents had total waste separation attitudes, but only 45% had waste separation behavior, which is a good starting basis for the following analysis. In total, knowledge in urban regions was higher than in rural regions, and there was no difference regarding attitudes or behavior. In addition, in urban regions, knowledge and attitudes were not different. Waste separation behavior in a pilot city was more frequent than in a non-pilot city.

For knowledge scores, the right ratio of recyclable waste between rural and urban locations and between a non-pilot city and a pilot city was not different, and they were all low at only 0.17, but the right ratio of harmful waste between rural and urban regions was significantly different at 0.29 and 0.37, respectively. There was no significant difference between pilot and non-pilot cities in urban regions, but the knowledge of residents in a pilot city was higher than in a non-pilot city. Regarding attitude scores, the willingness to separate, participate, and pay was not different at approximately 0.7 to 0.97, which means that the residents had a stronger intention to separate waste. To measure practices, we used the waste separation actions in daily life as waste separation behavior, which was 0.45. The behavior in rural and urban regions was not different, but there was a gap of 0.14 between pilot and non-pilot cities.

We also surveyed waste separation habits as a substitute for waste separation behavior and the difficulty of separating waste as a substitute for waste separation knowledge and attitudes for testing the stability of the results. We also found that the difficulty of separating waste did not differ among regions and cities. The habits between rural and urban regions showed no difference, but there was a significant difference between the pilot and non-pilot cities, which scored 0.6 and 0.69, respectively.

At the same time, we also compared cultural consumption among regions and cities, and we found that non-pilot and pilot cities were almost at the same level with regard to all types of cultural consumption. In rural and urban regions, there was also no difference regarding mass cultural consumption. But with regard to elite cultural consumption, that of residents in urban regions was significantly higher than it was in rural regions, especially regarding the cultural consumption of museums and cultural sites.

### 3.2. Correlations between Cultural Consumption and Knowledge, Attitudes, and Practices toward Waste Separation

For a better description of the relationship between cultural consumption and the KAPs toward waste separation, we designed Appendix A Table A3. In total, we found that mass cultural consumption had a positive impact on waste separation attitudes but no impact on waste separation knowledge and behavior. Elite cultural consumption had a negative impact on knowledge and attitudes but had a positive impact on behavior and habits.

In singling out cultural consumption, we found that when residents watched the news, their knowledge and attitude scores were higher; when residents read books, their attitude scores were higher; when they watched movies, their knowledge was lower, but the possibility of reporting the difficulty in separating waste decreased to 0.79. Interestingly, only movies had a negative impact on behavior and habits at 0.11 and 0.09, respectively. This result is also verified by several studies on movies that are classified as mass cultural consumption and always depict materialism through the lens of an individual’s value [27,29,51,53]. In practice, most movies are over-commercialized in China and do not sufficiently mirror important social and environmental questions. When residents visited museums, their knowledge and attitudes decreased, and their waste separation behavior and habits increased. When residents went to see dramas, their behavior and habits also increased. Visiting cultural sites also had the same impact on habits but no impact on behavior.

### 3.3. Impact of Cultural Consumption on Knowledge, Attitudes, and Practices of Waste Separation

We conducted a multi-variable regression to analyze the impacts shown in Table 3, the results of which are similar to the descriptive analysis in Appendix A Table A2. Models 1–3 and Models 4–6, respectively, show the relationships among a single type of cultural consumption, community waste separation services, socioeconomic characteristics, and knowledge, attitudes, and practices of waste separation in total and only in urban regions. Then, we used Models 7–9 and Models 10–12 to analyze the impacts of mass and elite cultural consumption on the KAPs of waste separation with the control of community separation services and socioeconomic characteristics in all regions and urban regions only. In total, the results were similar between all regions and urban regions.

Models 1–3 show the impact of cultural consumption across all regions. In the single cultural consumption case, news and books could significantly increase the knowledge and attitudes toward waste separation, but it did not impact on behavior; movies could significantly decrease knowledge and behavior, but it did not impact on attitudes toward waste separation. However, regarding elite cultural consumption, visiting museums decreased knowledge and attitudes toward waste separation; dramas decreased waste separation attitudes, but they did not have an impact on behavior. Models 4–6 show that the results in urban regions were better than in rural regions. When the possibility of seeing the news increased by 1%, knowledge and attitudes scores significantly increased by 11.2% and 10.5%, respectively. However, the result regarding movie consumption was contrary to the previous results in Italy [27,29]. When the possibility of going to the cinema increased by 1%, knowledge and behaviors decreased by 9.4% and 12.9%, respectively. Regarding elite cultural consumption, we found that cultural consumption had no impact on knowledge of waste separation, but if residents went to the museum or saw dramas, attitudes toward waste separation decreased. In addition, only for residents who visited culture sites did the possibility of separating waste increase by 7.9%.

Models 7–9 show the overall impacts of mass and elite cultural consumption in total regions. The results show that based on the total sample, mass cultural consumption improved attitudes toward waste separation but decreased related behavior. Regarding elite cultural consumption, it decreased attitudes and increased behavior. When residents across all total regions increased their mass and elite cultural consumption by 1%, their attitudes increased by 8.3% and decreased by 5.1%. On the contrary, behavior decreased by 6.3% and increased by 6.3%. Models 10–12 show that when the possibility of engaging in mass cultural consumption increased by 1% in urban regions, attitudes increased by 7.6% and behavior decreased by 10.2%. When the possibility of engaging in elite cultural consumption increased by 1%, knowledge and attitudes decreased by 4% and 5.3%, respectively, but they had no impact on behavior.

In addition, we found that regarding community waste separation services, only waste separation facilities significantly increased the KAPs of waste separation, waste advertisements decreased knowledge and attitudes of waste separation and increased the behavior of waste separation, while waste rules only significantly impacted on waste separation behavior.

Regarding socioeconomic characteristics, the main factor was the age and health of the residents and their participation in environmental activities. Older residents had higher knowledge and attitudes. Residents with better health had higher attitudes and behavior. But in the total sample, female residents had better attitudes toward separating waste, and residents with higher incomes had a higher possibility of separating waste, but residents with a higher education decreased the ratio of waste separation. In addition, higher participation in environmental activities led to stronger attitudes and increased behavior. In urban regions, female residents had better knowledge and attitudes, but environmental activities only affected attitudes, and income and education both had no impact on attitude and behavior. We also found that education could decrease the ratio of waste separation.

### 3.4. Robustness Test

Based on the behavior of waste separation, we used the probit model to conduct a robustness test as shown in Appendix A Table A3. We used the difficulty of waste separation to substitute knowledge and attitudes because of the precondition of knowledge and attitudes to change one’s behavior. We used waste separation habits as the behavior over the long run. We found that the results were consistent with the results reported in Table 3. Reading the news could increase the possibility of experiencing difficulty in waste separation by 29.7% in urban regions. Watching movies increased the possibility of having difficulty in separating by 22.8% and 29.1% in all regions and urban regions, respectively, and it separately decreased the possibility of separation behavior and habits in the long run by approximately 36.7% and 35.7% across all regions. In urban regions, the result was the same for all regions.

When separating the cultural factors into two types of cultural consumption, we found that mass cultural consumption had a positive impact on the difficulty of waste separation but had a negative impact on separation behaviors. However, elite cultural consumption had a positive impact on both behavior and habits. These results were the same as those shown in Table 3. Thus, we believe our results are robust.

## 4. Discussion

This paper analyzed the impacts of cultural consumption on residents’ waste separation knowledge, attitudes, and practices, and, particularly, their waste separation behavior and habits. First, the data indicated that residents’ waste separation knowledge score was only 0.24, but their attitude score was 0.97, while their behavior score was 0.48. Second, differences between rural and urban regions existed regarding knowledge and behavior, but there was no difference regarding attitudes. Third, based on the results of the descriptive and regression analysis, we found that mass and elite cultural consumption had different impacts on knowledge, attitudes, and practices.

Mass cultural consumption can improve knowledge and attitudes but decrease practices. In mass cultural consumption, news and books had the most positive impact on knowledge and attitudes but going to the cinema had a negative impact on behavior. A possible explanation includes two aspects: On the one hand, news and books represent high-frequency mass cultural consumption, which facilitates access to information acquisition, increasing pro-social and pro-environmental knowledge; thus, news and books can increase knowledge and attitudes [46]. On the other hand, many movies, as over-commercialized culture goods, always include a context that encourages materialism and emphasizes wealth and socioeconomic status, which affects a utilitarian society [51]. Based on Bourdieu’s cultural sociology theory and Karl’s consumerism sociology theory, mass cultural consumption, such as watching the news and going to the cinema, can communicate materialism, in which the context is always the current society and is counter to public social expectations based on extrinsic motivations, so they cannot increase intrinsic motivation and affect waste separation behavior [47,51,52,58].

Elite cultural consumption could improve behavior and habits in urban regions but not knowledge and attitudes. These results are consistent with previous results [45]. Going to a museum and seeing dramas had a negative impact on knowledge and attitudes but going to culture sites had a positive impact on behavior. A possible explanation includes two aspects: on the one hand, based on the Bourdieu’s cultural sociology theory and Karl’s consumerism sociology theory, elite cultural consumption, such as going to museums, seeing dramas, and visiting culture sites, reflect ideas and interactions that guide the actions of individuals and provide intrinsic motivation for caring about social and environmental questions; thus, such behavior increases [17,46,57,58]. On the other hand, museums and dramas related to environmental issues are almost non-existent; therefore, individuals with a higher cultural consumption of museums and dramas have informational access to knowledge and attitudes about waste separation with the limit of time and capital. Thus, elite cultural consumption cannot provide detailed information, as does mass cultural consumption, and shows in-depth content behind social and environmental questions over the long run [17].

Based on the regression results, between the rural and urban regions and the pilot and non-pilot cities, there was no impact on the KAPs of waste separation, which was caused by other factors such as cultural consumption, personal characteristics, knowledge, attitudes, and community services. In addition, we found that regarding community waste services, waste separation facilities and rules had a positive impact on behavior, but waste advertising had a negative impact on knowledge and a positive impact on behavior. The reason for this may be that advertising in the community is always about the meaning and methods of waste separation using complex methods and texts instead of using simple and interesting methods and texts, so that the increase in knowledge was relatively low. In addition, we found that attitudes toward waste separation positively impacted knowledge at the 1% level. However, there was no significant impact on behavior, suggesting cultural consumption had the largest impact on the results.

There were several limitations to this study. First, this was a cross-sectional study and was, thus, unable to identify causal relationships. Our results are merely correlations. Secondly, we used seed investigators to conduct the questionnaire; therefore, the sample of residents was not a nationally representative sample. Thirdly, there may be selection and reporting bias. We did not use random sampling, so the results may have selection bias. Moreover, the knowledge, attitudes, and behaviors were the reported results; thus, a reporting bias existed. Fourthly, even though the results were similar to those of the Italian municipal waste management study [14,45], more studies are needed on the transferability to other countries due to the fact of cultural differences in different counties.

## 5. Conclusions

Based on the results, we found that waste separation knowledge was very low but that attitudes toward it were relatively more positive, and half of all surveyed residents engaged in waste separation behavior. In addition, there was no difference regarding KAPs among rural and urban regions, pilot cities, and non-pilot cities. Cultural consumption had different impacts on residents’ KAPs toward waste separation, and mass and elite cultural consumption had contrary effects. Mass cultural consumption can improve knowledge and attitudes but not practices. However, elite cultural consumption can increase attitudes and behavior. When we used the difficulty of waste separation as a substitute for knowledge and attitudes and waste separation habits as a behavior in the long term, the results were always same.

Based on the results, our suggestions for waste management are as follows: (1) although individuals’ attitudes and behavior in China are higher than other studies’ results, the government and communities should continue to increase residents’ knowledge of waste separation collection and increase accurate waste separation behavior and habits; (2) the government should invest in the construction of cultural facilities and carry out elite cultural activities based on the local culture in both rural and urban regions that focus on environmental issues or events, for example writing a drama or building a museum or exhibition center covering waste separation; (3) society should be encouraged to form a culture of collective behavior for a beautiful homeland, e.g., building town halls or inter-community development associations that can provide guidance on institutional legislation, directing the improvement of residential structures of mass and elite cultural consumption, and seeking appropriate properties for comprehensively increasing KAPs; (4) the government or non-profit organizations should encourage and support younger generations to gain expertise in developing a professional career in waste management and to reasonably practice cultural consumption.

In the future, studies should be expanded beyond these four aspects: one is to seek a basic cultural conception from Eastern culture or traditional culture in China, which is a possibility for solving waste separation collection at the root of the problem. In addition, by using different sociocultural backgrounds, such studies can compare the differences between Eastern and Western cultures in the European Union and pioneering countries. Second is to seek effective methods of solving waste separation from the perspective of representatives of institutions, who are the first gatekeepers in waste services provision, for example, investigate their level of education and familiarization and awareness of waste separation management for better waste management and guidance. Third is calculating the cost and benefit of different cultural consumptions for effective results using national data and individual microdata, which can provide cultural facilities and activities investment. Fourth is continuing to explore the mutual impact of waste management on the health level of a population and the change of waste management methods in the upcoming aging of societies.

## Figures and Tables

**Table 1 ijerph-19-00338-t001:** Descriptive statistics.

Variables	Value	Number	Mean	SD
**Socioeconomic characteristic**				
Gender	Female	681	0.57	0.49
	Male	508	0.43	0.49
Age	Lower than 24	783	0.66	0.47
	25–34	135	0.11	0.32
	35–44	129	0.11	0.31
	45–54	122	0.10	0.30
	More than 54	20	0.02	0.13
Education	Junior high school or lower	59	0.05	0.22
	High/technical secondary school	100	0.08	0.28
	Bachelor’s degree	870	0.73	0.44
	Master’s degree or higher	160	0.13	0.34
Income (USD) per month	Lower than 782	527	0.44	0.50
	782 to 1564	372	0.31	0.46
	1564 to 2346	143	0.12	0.33
	More than 2346	147	0.12	0.33
Health	Not good	40	0.03	0.18
	General	335	0.28	0.45
	Good	485	0.41	0.49
	Very good	329	0.28	0.45
Environmental activities	No	830	0.70	0.46
	Yes	359	0.30	0.46
**Community waste services**				
Community waste separation management advertisement	No	429	0.36	0.48
	Yes	760	0.64	0.48
Community waste separation management rule	No	521	0.44	0.50
	Yes	668	0.56	0.50
Waste separation collection facilities	No	853	0.72	0.45
	Yes	336	0.28	0.45
**Community location**				
Regions	Rural	354	0.30	0.46
	Urban	835	0.70	0.46
Pilot city	Non-pilot city	426	0.51	0.50
	Pilot city	409	0.49	0.50

Source: Authors’ survey.

**Table 2 ijerph-19-00338-t002:** Knowledge, attitudes, and practices of waste separation and cultural consumption across different regions in China.

Mean	Total	Rural	Urban	Rural and Urban ^5^	Non-Pilot City	Pilot City	Non-Pilot and Pilot City ^5^
**Waste separation knowledge ^1^**	0.26	0.22	0.27	−0.048 *	0.25	0.3	−0.048
Right recyclable waste	0.17	0.16	0.17	−0.013	0.15	0.19	−0.038
Right harmful waste	0.34	0.29	0.37	−0.084 **	0.34	0.4	−0.058
**Waste separation attitudes ^2^**	0.83	0.84	0.83	0.011	0.82	0.84	−0.021
Willingness to separate waste	0.7	0.71	0.7	0.003	0.68	0.73	−0.055
Willingness to separate with house separation facilities	0.8	0.81	0.80	0.006	0.79	0.82	−0.028
Willingness to separate with community separation facilities	0.97	0.97	0.97	−0.005	0.97	0.98	−0.004
Willingness to participate in community waste activities	0.75	0.77	0.74	0.034	0.74	0.74	−0.006
Willingness to pay for waste	0.93	0.94	0.93	0.017	0.92	0.93	−0.014
Difficulty in separating	0.77	0.79	0.77	0.023	0.77	0.76	0.005
**Waste separation practices**							
Waste separation behavior	0.45	0.42	0.47	−0.050	0.4	0.54	−0.141 ***
Waste separation habit	0.64	0.64	0.64	0.001	0.6	0.69	−0.091 **
**Mass cultural consumption ^3^**	0.78	0.78	0.78	−0.005	0.77	0.79	−0.015
News	0.91	0.89	0.92	−0.032	0.92	0.92	0.001
Books	0.93	0.93	0.93	0.000	0.92	0.94	−0.019
Movies	0.87	0.88	0.87	0.007	0.86	0.88	−0.023
**Elite cultural consumption ^4^**	0.38	0.33	0.39	−0.06	0.41	0.38	0.032
Museums	0.62	0.58	0.64	−0.063 *	0.62	0.66	−0.031
Dramas	0.46	0.43	0.47	−0.047	0.50	0.44	0.060
Culture sites	0.68	0.62	0.71	−0.098 ***	0.73	0.70	0.033

Note: ^1^ Waste separation knowledge is the average of recyclable waste and harmful waste properly. ^2^ Waste separation attitude is the average of five willingness, including Willingness to separate waste, Willingness to separate with house separation facilities, Willingness to separate with community separation facilities, Willingness to participate in community waste activities and Willingness to pay for waste. ^3^ Mass cultural consumption was a binary variable, which was 1 if residents consumed news, books, or movies. ^4^ Elite cultural consumption was also a binary variable, which was 1 if residents attended museums, dramas, or culture sites. ^5^ These is the difference between rural and urban regions, Non-pilot and pilot city. *** *p* < 0.01, ** *p* < 0.05, * *p* < 0.1.

**Table 3 ijerph-19-00338-t003:** Impact of cultural consumption on waste separation knowledge, attitudes, and practices.

Sample	Total Sample			Urban Sample			Total Sample			Urban Sample		
Variables	Knowledge	Attitude	Behavior	Knowledge	Attitude	Behavior	Knowledge	Attitude	Behavior	Knowledge	Attitude	Behavior
	(1)	(2)	(3)	(4)	(5)	(6)	(7)	(8)	(9)	(10)	(11)	(12)
**Cultural consumption**												
News	0.093 ***	0.112 ***	0.016	0.114 ***	0.109 ***	−0.057						
	(0.029)	(0.025)	(0.058)	(0.034)	(0.032)	(0.067)						
Books	0.092 ***	0.130 ***	−0.007	0.073	0.129 ***	−0.062						
	(0.032)	(0.030)	(0.052)	(0.048)	(0.040)	(0.069)						
Movies	−0.103 *	0.024	−0.125 ***	−0.094 *	0.034	−0.128 ***						
	(0.051)	(0.018)	(0.029)	(0.055)	(0.025)	(0.039)						
Museums	−0.044 *	−0.044 ***	0.019	−0.039	−0.046 **	0.028						
	(0.023)	(0.015)	(0.031)	(0.028)	(0.020)	(0.038)						
Dramas	−0.003	−0.026 **	0.030	0.004	−0.024	−0.037						
	(0.018)	(0.013)	(0.034)	(0.023)	(0.016)	(0.042)						
Culture sites	−0.023	0.018	0.030	−0.031	0.003	0.080 **						
	(0.030)	(0.012)	(0.031)	(0.042)	(0.013)	(0.034)						
Mass cultural							−0.012	0.083 ***	−0.061 **	0.000	0.076 ***	−0.101 ***
Consumption							(0.031)	(0.012)	(0.027)	(0.029)	(0.016)	(0.034)
Elite cultural							−0.037	−0.051 ***	0.063 **	−0.041 *	−0.054 ***	0.033
Consumption							(0.029)	(0.014)	(0.023)	(0.023)	(0.019)	(0.028)
**Waste knowledge and attitudes**												
Waste separation knowledge		0.059 ***	0.024		0.057 ***	0.073		0.069 ***	0.030		0.065 ***	0.073 *
		(0.016)	(0.034)		(0.020)	(0.045)		(0.017)	(0.031)		(0.020)	(0.039)
Waste separation			0.100			0.102			0.109			0.096
attitude			(0.089)			(0.106)			(0.086)			(0.103)
**Community waste services**												
Waste separation	0.066 *	0.025	0.080 **	0.079 **	0.031	0.052	0.070 *	0.028 *	0.079 **	0.087 **	0.035	0.054
facility	(0.033)	(0.015)	(0.038)	(0.038)	(0.025)	(0.037)	(0.036)	(0.015)	(0.039)	(0.040)	(0.026)	(0.039)
Waste advertising	−0.041 *	−0.011	0.178 ***	−0.051 *	−0.007	0.226 ***	−0.035	−0.008	0.182 ***	−0.049 *	−0.006	0.227 ***
	(0.023)	(0.011)	(0.031)	(0.025)	(0.014)	(0.029)	(0.024)	(0.010)	(0.030)	(0.024)	(0.013)	(0.029)
Waste rules	0.007	0.011	0.192 ***	0.025	0.001	0.213 ***	0.001	0.007	0.190 ***	0.022	−0.001	0.212 ***
	(0.017)	(0.015)	(0.037)	(0.018)	(0.026)	(0.042)	(0.016)	(0.017)	(0.038)	(0.016)	(0.027)	(0.045)
**Socioeconomics characteristics**												
Gender of residents	0.021	0.088 ***	0.009	0.032 **	0.093 ***	−0.008	0.018	0.088 ***	0.005	0.030 *	0.094 ***	−0.009
	(0.017)	(0.015)	(0.021)	(0.015)	(0.016)	(0.026)	(0.018)	(0.015)	(0.020)	(0.015)	(0.015)	(0.026)
Age of residents	0.067 ***	0.026 ***	−0.009	0.054 ***	0.033 **	−0.013	0.080 ***	0.030 **	−0.000	0.067 ***	0.036 **	−0.006
	(0.011)	(0.009)	(0.019)	(0.010)	(0.012)	(0.020)	(0.010)	(0.011)	(0.019)	(0.008)	(0.014)	(0.020)
Edu of residents	−0.003	−0.006	−0.028 *	−0.006	0.007	−0.022	−0.004	−0.003	−0.029 **	−0.010	0.010	−0.024
	(0.012)	(0.009)	(0.015)	(0.015)	(0.014)	(0.016)	(0.013)	(0.009)	(0.013)	(0.017)	(0.013)	(0.016)
Income of residents	0.008	−0.006	0.031 **	0.009	−0.005	0.029	0.008	−0.006	0.031 **	0.010	−0.004	0.031
	(0.009)	(0.010)	(0.015)	(0.011)	(0.009)	(0.020)	(0.010)	(0.010)	(0.014)	(0.011)	(0.009)	(0.020)
Health	0.012	0.027 ***	0.028 *	0.012	0.029 **	0.036 *	0.011	0.026 ***	0.028 *	0.010	0.028 **	0.037 *
	(0.010)	(0.008)	(0.016)	(0.011)	(0.014)	(0.019)	(0.011)	(0.007)	(0.016)	(0.012)	(0.013)	(0.018)
Environment activity	0.029	0.048 ***	0.066 **	0.043	0.061 ***	0.042	0.034	0.054 ***	0.065 **	0.046	0.065 ***	0.041
	(0.034)	(0.012)	(0.026)	(0.036)	(0.018)	(0.038)	(0.034)	(0.010)	(0.026)	(0.034)	(0.017)	(0.034)
Place of residents	0.021	−0.018	−0.005				0.019	−0.017	−0.004			
	(0.030)	(0.018)	(0.026)				(0.028)	(0.019)	(0.027)			
Pilot city				0.023	−0.005	0.017				0.018	−0.007	0.015
				(0.020)	(0.019)	(0.032)				(0.020)	(0.019)	(0.038)
Constant	0.058	0.483 ***	0.137	0.070	0.413 ***	0.177	0.107	0.632 ***	0.086	0.125	0.568 ***	0.077
	(0.101)	(0.044)	(0.093)	(0.112)	(0.096)	(0.115)	(0.075)	(0.036)	(0.091)	(0.083)	(0.087)	(0.116)
Observations	1189	1189	1189	835	835	835	1189	1189	1189	835	835	835
*R*-squared	0.065	0.126	0.159	0.068	0.134	0.191	0.048	0.103	0.156	0.052	0.111	0.185

Note: Robust standard errors in parentheses. *** *p* < 0.01, ** *p* < 0.05, * *p* < 0.1.

## Data Availability

The data presented in this study are available upon request from the corresponding authors.

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
