# Peer review of "Cultural Consumption and Knowledge, Attitudes, and Practices Regarding Waste Separation Management in China"

_ijerph, 2021, doi:10.3390/ijerph19010338_

Round 1

Reviewer 1 Report

The paper is interesting, logically structured, important through the relationship established with environmental pollution issues and actual for the protection of public health, both in China and in any other country in the world.

The introduction provides information on the importance of the topic and establishes the context, but also the evolution of regulations for the problem that the research aims at. The purpose and structure are formulated here and provide a clear picture of the authors' approach. Establishing a correlation between personal motivations - community culture (with a set of rules and regulations more or less formally regulated), as well as the triad of knowledge, attitudes, behaviors is particularly interesting, especially transposed in the issue of public policy and waste management. My suggestion is that the authors emphasizing the elements of originality of their work.

The theoretical framework is established on the basis of the selected theory, but does not highlight the results and / or limitations of previous models, nor does it bring arguments regarding their differentiation. I suggest a firmer/stronger position towards what is already known or established both conceptually and as methods or implementations of management systems, in order not to offer the opportunity to diminish the value of research work.

I suggest whether it is possible to make a brief check on aspects related to people's health in rural or urban areas that could also impact attitudes and reveal important social behaviors. A paragraph on economic issues could be introduced, e.g. the costs involved in behaviors, or what is needed to finance cultural consumption for effective results, important aspects for public policy decisions.

In point 2, in particular 2.2, I would suggest a brief explanation of the decision to choose the model and the sample of respondents, as well as the impact of their remuneration.

The results are explained, but I even suggest an explicit association of results similar or contrary to those in the paper, in order to clarify the position of the authors.

The conclusions are logical and clear, but quite brief. In order to raise the level of knowledge and change attitudes and behaviors, the authors suggest that the government provide facilities or carry out activities. Could the authors themselves propose solutions for the government, distinct for urban and rural regions, with the anticipation of effects?

It would also, be interesting to know the level of education, familiarization and awareness among the representatives of the institutions responsible for waste management or regulatory trends in the field. Would the results of the research brought to the attention of these institutions contribute to the guidance of those responsible for the application of the legislative provisions (town halls or inter-community development associations)?

Could the study contribute to providing a public policy proposal to support the younger generation in gaining the expertise needed to develop a professional career in waste management? Could research continue on the impact of waste management on the health level of the population? Is it a research topic with a future outlook in China?

I think the results already exposed allow to develop the ideas in many different ways.

Congratulations on your work and good luck!

Author Response

We first would like to thank the reviewer for his/her careful review of our paper. We believe that these comments and suggestion were beneficial in revising the manuscript. We have tried to respond to all of them in full. In order to facilitate your re-review of these issues, we have included with this letter a detailed, comment-by-comment response for each section of the manuscript. Here, we first reproduce the comments of the reviewer, and then in our indented paragraphs we provide a summary of our response. Newly added sentences in the revised manuscript are then shown in red italics.

Reviewer 2 Report

Please update as per the attached file of my comments.

Reviewer 3 Report

The results of social empirical research from the People's Republic of China can be used beneficially there, but also in many other countries. However, the authors themselves do not clarify the possibilities and limits of transferability to other countries or even a generalizability.

Regarding knowledge, it is unfortunately not clear whether "right (?) waste separation knowledge" differs from "adequate waste separation know-how", i.e. more precise insights into the difference between knowledge and know-how are missing here, but the question also arises how to distinguish "right" from "wrong" knowledge with regard to waste separation. It is well known from countries such as germany that the practical material knowledge of different groups of actors about material recycling is at odds with separation practices that run along socio-economic criteria rather than material criteria. In the packaging fraction in particular, a good material knowledge in Germany consequently leads to incorrect separation practice.

The differentiation achieved in parts 3 and 4 of the study is abandoned in the conclusion, when it is succinctly stated: "Our findings suggest that cultural consumption is one of the most efficient methods for improving waste separation knowledge, attitudes, and behavior." If the intention is "to increase cultural consumption," this could not only be a task for the government. It could also be for interested authors and journalists to develop new offerings that overcome the detected weaknesses of existing offerings (e.g. writing a drama covering waste separation).
Recommendation:
The paper is essentially not built on a (socio-)cultural understanding of how to deal with waste, but on a rather traditional motivational psychological concept of a waste behavior, which is primarily economically motivated and unfortunately does not allow a satisfactory outlook towards possibilities for change (beyond "adding cultural facilities and activities"). 
In my view, it is not credible for the presentation to continue to pretend that it is based on a cultural concept, since this concept is neither defined nor based on culture or science. It is probably not possible to deal adequately with this serious deficit without a change in the authors and without a change in the analysis of the empirical results. As a way out, the authors therefore could ask in the outlook what a cultural conception would have to look like in the People's Republic of China that would make it possible to cope with waste more appropriately than before; for such a question, it might also be possible to build on sociocultural studies from the European Union and the pioneering countries there.   
